# Hip Fractures and Visual Impairment: Is There a Cause–Consequence Mechanism?

**DOI:** 10.3390/jcm11143926

**Published:** 2022-07-06

**Authors:** Gianluca Testa, Sara De Salvo, Silvia Boscaglia, Marco Montemagno, Antonio Longo, Andrea Russo, Giuseppe Sessa, Vito Pavone

**Affiliations:** 1Department of General Surgery and Medical Surgical Specialties, Section of Orthopaedics and Traumatology, University Hospital Policlinico-San Marco, University of Catania, 95123 Catania, Italy; sarads94@hotmail.it (S.D.S.); docmontemagno@gmail.com (M.M.); giusessa@unict.it (G.S.); vitopavone@hotmail.com (V.P.); 2Department of General Surgery and Medical Surgical Specialties, Section of Ophthalmology, University Hospital Policlinico-San Marco, University of Catania, 95123 Catania, Italy; silviabosc@hotmail.it (S.B.); antlongo@unict.it (A.L.); andrea.russo@unict.it (A.R.)

**Keywords:** aged, cataracts, elderly, falls, femur fractures, hip fractures, sight defects, vision, visual acuity, visual impairment

## Abstract

Background: Numerous studies have pointed out how visual impairment relates to falls in the elderly, causing dangerous consequences, such as fractures. The proximal femur fracture is one of the most frequent fracture types related to poor vision. This study investigates the link between fall-related hip fractures and visual impairment. Methods: The present is an observational monocentric case–control study. We collected the ophthalmologic anamnesis and measured the visual acuity of 88 subjects with femur neck fracture (case group), comparing it with 101 adults without fractures and a recent fall history. Results: The results showed no statistical difference between the two groups regarding visual acuity, with a *p*-value of 0.08 for the right eye and 0.13 for the left one. One of the major ophthalmologic morbidities found was cataracts, present in 48% of the control group and 30% of the case group. Conclusions: The data obtained suggest that visual impairment might not be crucial in determining falls in the elderly.

## 1. Introduction

As the population ages and becomes more fragile, the rate of falls having significant health consequences rises, creating a worldwide phenomenon. Fractures are one of these consequences, with femur neck fractures being dangerous for their association with fitness depletion and mental disorders [1,2]. When possible, surgical treatment is the preferred choice [3], with various techniques and implants used based on the fracture pattern and the patient’s fitness. However, any surgery has risks, especially in the older population [4]. Particularly in well-developed countries, hip fractures are a significant cause of death and physical disability in the elderly, mainly in the postoperative period, representing a real health emergency. International and national guidelines suggest treating these patients as soon as possible, within 24–48 h, to avoid these serious consequences [5].

Why do old adults fall? Many researchers have attempted to answer this question by addressing different explanations. Among these, there is visual impairment: growing old means losing visual acuity [6]; therefore, it seems logical to expect higher chances of accidental falls. Moreover, osteoporosis, sarcopenia [7,8], arthrosis, dementia, and balance problems [9] make falls far more likely in those aged over 65 [1]. Numerous studies in the scientific literature show how visual impairment in elderly patients seems connected to a higher fall rate, often resulting in fractures, especially hip ones [10]. The present study aims to investigate the matter further. It adds different data compared to the previous literature, because it shows no significant correlation between visual impairment, falls, and femur neck fractures, widening the debate. 

## 2. Materials and Methods

We conducted an observational monocentric study with a case–control design to compare patients’ visual acuity affected by a femoral neck fracture and a cohort of randomly selected subjects.

From July 2020 to December 2021, we collected 88 patients with FNF, surgically treated at the orthopedic clinic of the University of Catania, enrolled after Ethical Committee approval. The patients admitted through the emergency department were grouped from medical records based on the following personal data: gender, age at the time of trauma, fall mechanism, fracture type. The patients selected for the study were those with femoral neck fractures classified as AO/OTA: 31A–31B, over 65 years old at the time of trauma, and referring only to accidental falls as the cause of the fractures. They were subjected to an ophthalmic examination comprehending the use of the optotype and a thorough ophthalmic anamnesis.

The control group was created by randomly contacting 101 adults aged over 65, offering them a free ophthalmologic exam. Only patients without a history of FNF and who had not experienced accidental falls in the past six months were selected. Once they reached the hospital, they were given the same exam as the case group, such as eye chart test and ophthalmic anamnesis.

We assessed 189 adults’ visual acuity, for a total of 378 eyes. The results between the two groups were statistically analyzed with the *t*-Student test.

Moreover, we analyzed the odds ratio (OR) of the fracture population (case group), considering as cut-off 3/10 of visual acuity, as used to indicate visual impairment globally (Figure 1). According to the Italian law, in fact, mild visual impairment is attributed to those who have a visual acuity no greater than 3/10 in both eyes or in the best eye, even with the best correction, and those who have a peripheral binocular visual field residual lower than 60% [11].

The case group’s mean age was 84, ranging from 66 to 99. The control group’s mean age was 76 (range 66–93). Despite the age gap, the visual acuity difference in these groups was not statistically significant, with a *p*-value of 0.09 for the right eye and 0.13 for the left one.

The odds ratio gave similar results, considering VO < 0.3 as a cut-off for the study. In fact, the OR = 0.54 with C.I. (95%) = 0.71 ± 0.05. Both measures showed no correlation between visual impairment (intended as VO < 0.3) and femur neck fractures. 

One of the major ophthalmologic morbidities found was cataracts, present in 48% of the control group and 30% of the case group. Other ophthalmic conditions were glaucoma, maculopathies, and tumors. The presence of these diseases was not numerically predominant if compared to the statistical sample. The only condition that we found more frequent in the FNF group was glaucoma; the others were all more frequent in the control group. Maculopathies and cancer were found mainly in the control sample. The detailed data are displayed in Table 1.

## 3. Discussion

Hip fractures are a major public health problem, especially in western countries. As one of the most frequent injuries in the elderly (over 65 years of age), their incidence is still rising worldwide: they are expected to reach over 6 million by 2050 [3].

This condition accounts for a high morbidity and mortality rate; throughout the years, different techniques have been developed for its treatment [12].

Conservative therapy is not advisable since it lowers the quality of life and the overall survival rate of the patients; therefore, surgery is the treatment of choice for most cases, allowing early rehabilitation and functional recovery [13]. Over time, different systems have been developed, and surgeons choose based on the fracture pattern and patients’ fitness. From intramedullary nailing to hip arthroplasty, research nowadays focuses on improving mechanics and materials for the best outcomes in terms of quality and duration [14]. Hip surgery is still a high-risk surgery, especially for the elderly patients who undergo these types of procedures.

Amongst the most frequent causes of hip fracture, it is essential to cite visual impairment, osteoporosis, sarcopenia, dementia, and all other factors that make elderly subjects particularly fragile. Sight defects belong to this panorama, often neglected by patients that do not have easy access to prevention and healing facilities. Several works in the literature have tried to measure this association, with the first ones being conducted in the 1970s [15]. The majority showed some correlation, but, of course, every study has its limits and biases [16].

In fact, previous studies have used different methods to evaluate visual impairment, trying to obtain accurate data. Some used questionnaires to assess visual acuity, giving a picture of patients’ perceptions rather than objective testing [17]; others studied large health insurance databases [2] to provide the highest number of cases possible. We could not find a study in the past literature on falls and sight defects that does not evidence a correlation between the two. 

The data displayed for the present study were not expected. Moreover, given the different mean ages between the groups, we expected to find better visual acuity in the control group. Ageing is a risk factor for worse visual acuity [18]. When analyzing subjects’ anamnesis, we found a higher rate of cataracts in the control group, where the subjects’ mean age was 76 years. The case group instead had a mean age of almost 84 years. This study’s results stimulate further reflections and research due to the absence of a significant difference in the groups studied. The cause–effect correlation between sight problems and falls is unclear; falls resulting in hip fractures might not be directly correlated to sight problems. It is vital to investigate how the event happened; every fall described as accidental has its peculiarities that only a scrupulous anamnesis of the patient can capture. The location in which it happened, the trigger situation, the patient’s response, and their reflexes and coordination are elements that do not depend on eyesight only. Other factors probably influence these episodes more, as some studies have shown [19], such as balance disorders, gait problems, pharmaceutical use, and a history of falls. The most cited risk factor for falls is fall history [20], which indicates that patients are prone to falling, but the root of the problem remains unknown and is probably multifactorial. Regarding elderly patients, it is important to take into account the use of many different pharmaceuticals [21], often related to falls, since some active substances can alter normal reflexes and coordination. The presence of gait disorders [22] before the event, often underdiagnosed and undertreated, is another cause. Finally, fractures after these incidents happen in frail patients, and osteoporosis [23] is one of the main comorbidities associated with increasing the likelihood of fractures as a consequence of a fall.

Prevention is vital, but defining the causes of falls is essential, especially in senior patients. Since eyesight is not so reliable, other factors should be considered first. Patients with fall history should receive physicians’ attention, and, after determining the main risk factors, they should be guided towards prevention strategies. Coordination and balance disorders are frequently linked to falls; recent studies have suggested that preventive physiotherapy [24] might be vital in preventing falls. Preventive strategies should also include the fall’s consequences: a proper osteoporosis diagnosis and treatment can be very helpful in avoiding fractures [25].

The present study has several limitations. The collection of the control group might have been biased: subjects contacted for the ophthalmic examination might have been interested in participating due to the presence of sight defects, which they were suddenly given the opportunity to receive treatment for. Another limitation is the sample size, comparable to similar studies in Europe [26], being a monocentric study with 12 months of data collection.

Moreover, the case group could have been larger. In fact, we included a specific type of femur fracture, without considering other patients affected by different types of femur neck fractures with similar accidental traumatic events. 

## 4. Conclusions

Amidst the risk factors correlated to accidental falls in the elderly, visual impairment has been considered one of the most important in the current literature. In contrast with this vision, the data displayed in this article underline how there are no statistically significant differences in visual acuity between fractured and non-fractured patients. Therefore, the present study questions past knowledge, underlining how sight defects might not play a critical role in this condition. For this reason, further studies are needed to investigate this matter.

## Figures and Tables

**Figure 1 jcm-11-03926-f001:**
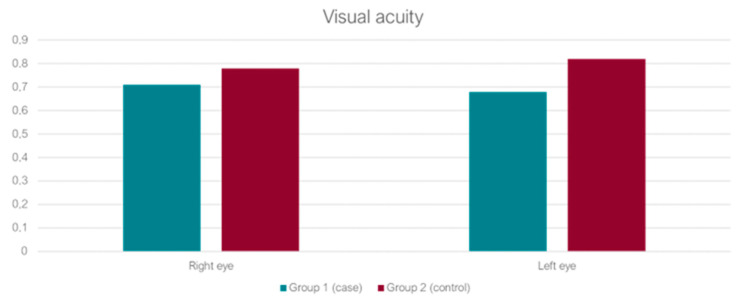
Visual acuity.

**Table 1 jcm-11-03926-t001:** Study statistics.

	Case Group	Control Group	*p*-Value
Patients (*n*)	**88** **M/F = 0.51**	**101** **M/F = 1.14**	
Mean age	83.9 R: 66–99	76 R: 66–93	*p* = 0.15
Visual acuity mean right (std)	0.71	0.78	*p* = 0.08
	(±0.20)	(±0.82)	
Visual acuity mean left (std)	0.68	0.82	*p* = 0.13
	(±0.22)	(±0.80)	
Cataract (%)	27	49	
	30%	48%	

## Data Availability

Study data are available from the corresponding author.

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
