# Peer review of "Hip Fractures and Visual Impairment: Is There a Cause–Consequence Mechanism?"

_jcm, 2022, doi:10.3390/jcm11143926_

Round 1

Reviewer 1 Report

1.      The uppercase and lowercase of the title should be revised according to MDPI format.

2.      Are there any quantitative results that would be added in the abstract section? It is recommended to add it.

3.      Keywords need to be reordered based on alphabetic order.

4.      To make the manuscript have more visibility, it is recommended to add one or more keywords.

5.      The present introduction is too short, it shows the low quality of the present manuscript. It is needed to be improved.

6.      What is the novel of the present article? Nothing really new has been presented in the present manuscript, lack of novel. The authors must be keen on this issue.

7.      Hip fracture is very dangerous and needed surgery with the total hip implant in specific cases. Please include this important point in the introduction/discussion section. Also, to support this explanation, it is recommended to adopt suggested references published by MDPI as follows: Computational Contact Pressure Prediction of CoCrMo, SS 316L and Ti6Al4V Femoral Head against UHMWPE Acetabular Cup under Gait Cycle. J. Funct. Biomater. 2022, 13, 64. https://doi.org/10.3390/jfb13020064

8.      Previous research with their findings and its limitation should be explained in at least one paragraph to show the raised research gaps.

9.      The study procedure needs to be included as an illustrative figure in the materials and methods section.

10.   The basis/procedure/protocols/standard in the present study is not clearly understood.

11.   Please arrange the paragraph with at least three sentences to make a solid explanation that consists of one main sentence followed by a supporting sentence. The paragraph in the present manuscript was found not in the correct way, for example in the last two paragraphs of the discussion section.

12.   The conclusion has not captured the study precisely; an overall rewrite is needed.

13.   Further study needs to be explained in the conclusion section.

14.   Overall, the present quality of the manuscript is poor and lacks content. Further improvement with massive additional substantial scientific contribution is really needed to make it suitable for publication.

15.   Please make sure the authors have followed the JCM temple perfectly. The authors can download the published version and compare it with the authors' manuscript to find not proper typesetting/format.

16.   The English language needs to be proofread to revise the error grammar and improve the English style.

Author Response

Dear Reviewer, thank you for your revision and suggestions to improve the article. Here you will find a point-by-point responses: 

  1. The uppercases and lowercases have been revised according to MDPI format.
  2. Quantitative results added in the abstract.
  3. Keywords are now in alphabetic order.
  4.  We added keywords.
  5. The introduction has been improved.
  6. The novel of this article is the inconsistency between the data displayed here and the data in the previous literature. In previous studies, there was a correlation between visual impairment and fractures. Here, there is not.
  7. Surgical treatment has been included in the introduction, surgical treatment of hip fracture has been described in the discussion, and we added citations too.
  8. The paragraph to explain the biases of previous studies has been added.
  9. An illustrative figure of the study has been added to the materials and methods.
  10. The article has been rewritten in some parts to make everything more understandable.
  11. The paragraphs have been rearranged.
  12. The conclusion has been rewritten.
  13. Further studies needed has been added.
  14. We added statistical data.
  15. The JCM template has been followed.
  16. The English language has been proofread.

Reviewer 2 Report

The current study 'Hip fractures and visual impairment: Is there a cause-consequence mechanism?', is a good attempt to understand the relationship between elderly patient's vision acuity and hip fractures. While this study seems like a great idea. Several studies in literature have been done to answer the same research question. For instance: Visual Impairment and Hip Fracture, Ivers et al., 2000; Visual impairment and hip fractures: a case-control study in elderly patients, Loriout et. al., 2014, are some studies that have shown an influence of visual acuity on fractured. However, the current investigation found no significant difference.

Some of the reasons for inconclusive results from the study are:

1. Poorly defined hypothesis: The statement given in the introduction is: ‘Numerous studies in the scientific literature show how visual impairment in elderly patients seems connected to a higher fall rate, often resulting in fractures, especially hip ones8. The present study aims to investigate the matter further, adding different data compared to the previous literature, widening the debate.’

This is a very vague statement and not a defined hypothesis. What is the data being added? How is this study adding anything to the knowledge base?

2. A poor experimental design follows an ill-defined hypothesis in this study.  The number of men and women in the test and control groups are clearly different. The internal subgroups are not defined, other than cataract, which is inadequate. Glaucoma for instance seems to be a major factor from some of the previous studies. Instead of doing subgroup analysis the authors have relied only on P values. It is suggested that the researchers go through literature to come up with a better study design. For instance using odds ratio (OR), and confidence interval (CI) over a simple comparison between the two groups is suggested. Taking some help from a statistician right from the inception of the study could avoid such problems too.

 3. Finally there is no acknowledgement of the uncertain results obtained. Instead statements with no citations to back them like ‘In general, eighty-year-old adults should have worse visual acuity than

seventy-year-olds with or without fall history.’ Are scientifically inaccurate random statements. These inaccurate random statements need to be eliminated from the manuscript.

4. The discussion is filled with all other factors not considered in the study. A quick review of literature would have informed the researchers the importance of these factors like osteoporosis and other visual disorders neglected in this study.

 In totality, the study does not add much to the knowledge base. It must be re-analyzed after formulation of a testable hypothesis and proper statistical methods.

Author Response

Dear Reviewer, thank you for your revision and suggestions to improve the article. Here you will find a point-by-point responses:

  1. The introduction has been changed by explaining why the data in the present study is different.
  2. We added the use of odds ratio to implement our statistics.
  3. The general statement has been substituted with a statement backed by literature.
  4. We added statistical data. The factors mentioned in the discussion are there to widen it and give a better picture of the situation. The study focuses only on visual impairment.

Round 2

Reviewer 1 Report

I am pleased to review this manuscript. It is improved from the submitted version.

Reviewer 2 Report

While much can not be done with a poorly thought out study, the authors tried their best to address the issues raised that could be resolved. However, it is important to think if the research question is worth asking, in the case of clinical research before anything else. We must keep that in mind even before starting a study.